# Influence of *Lactiplantibacillus plantarum* and *Saccharomyces cerevisiae* Individual and Collaborative Inoculation on Flavor Characteristics of Rose Fermented Beverage

**DOI:** 10.3390/foods14111868

**Published:** 2025-05-24

**Authors:** Yingjun Zhou, Yinying Chao, Chengzi Huang, Xiaochun Li, Zhuhu Yi, Zuohua Zhu, Li Yan, Yu Ding, Yuande Peng, Chunliang Xie

**Affiliations:** 1Institute of Bast Fiber Crops, Chinese Academy of Agricultural Sciences, Changsha 410205, China; zhouyingjun@caas.cn (Y.Z.); 17369242989@163.com (C.H.); li-xiao-chun@foxmail.com (X.L.); zhuzuohua@caas.cn (Z.Z.); yanli214@126.com (L.Y.); ibfcpyd313@126.com (Y.P.); 2College of Life Science, Hunan Normal University, Changsha 410181, China; chaoyinying2024@163.com; 3Zhangjiajie Agricultural Science and Technology Research Institute, Zhangjiajie 427000, China; yizhuhu@163.com; 4Department of Food Science and Engineering, College of Life Science and Technology, Jinan University, Guangzhou 510632, China; dingyu_cu@163.com; 5Yuelushan Laboratory, Changsha 410128, China

**Keywords:** rose beverages, co-fermentation, quality characteristics, volatile compounds, biomarkers

## Abstract

This study investigates the impact of using *Lactiplantibacillus plantarum* and *Saccharomyces cerevisiae*, either individually or in co-culture, on the fermentation of rose beverage. We comprehensively analyzed the resulting changes in quality characteristics and volatile compound profiles. Fermentation significantly altered the physicochemical properties, appearance, color, and free amino acid/organic acid content. Both microbial strains significantly increased total polyphenols and flavonoid content, with co-fermentation exhibiting a more pronounced effect compared to single-strain fermentations. Furthermore, the volatile compounds in rose beverages fermented with different microorganisms were characterized by an electronic nose (E-nose) and headspace–solid-phase microextraction coupled with gas chromatography–mass spectrometry (HS-SPME/GC-MS). E-nose analysis demonstrated distinct volatile profiles distinguishing the four fermentation samples. HS-SPME/GC-MS identified a total of 245 volatile compounds, among which alcohols constituted the most abundant class. Integrating GC-MS data with odor activity value (OAV ≥ 1) analysis pinpointed 34 key aroma compounds. Partial least-squares discriminant analysis (PLS-DA) based on variable importance in projection (VIP) identified eight key volatile markers: eugenol, phenylethyl alcohol, (E)-3,7-dimethyl-2,6-octadienoic acid, methyleugenol, ethyl octanoate, citronellol, D-citronellol, and 2,4-bis(1,1-dimethylethyl)phenol. These findings provide valuable insights into the microbial influence on rose beverage quality and offer a theoretical basis for optimizing industrial fermentation processes.

## 1. Introduction

The genus *Rosa* (Rosaceae) originates from Europe, Africa, and Asia and is widely cultivated in regions such as north and southwest China, Japan, and Korea [1]. Rose has a pleasant flavor and is a good source of vitamins C, amino acids, essential fatty acids, minerals, and other beneficial bioactive compounds [2,3]. Traditionally, roses have been utilized for their properties as aromatic agents, skin moisturizers, tonics for the spleen and stomach, uterine support, pain relief, and antioxidants [4]. Due to their high moisture content, fresh rose petals are highly perishable, necessitating processing into products such as jams, juices, cake ingredients, vinegar, teas, and flavored wines [5,6]. However, conventional processing methods of these products often result in inconsistent product quality and low economic value, posing significant challenges for the rose-processing industry. Therefore, exploring new processing technology to boost market competitiveness is key to the rose industry.

Driven by increasing health awareness, consumer demand for functional foods has risen, making their development a significant research focus in food science [7]. Among these, functional plant-based fermented beverages are experiencing rapid growth and represent a key segment of the food market. These beverages are typically produced by fermenting plant-derived substrates (such as fruits, vegetables, and mushrooms) using lactic acid bacteria or yeasts [8]. Functional plant-based fermented beverages overcome the inherent shortcomings of non-fermented dairy products, such as lactose intolerance, and can be used as a substitute for milk and other dairy products [9]. Fermentation is a traditional processing method originally used for food preservation and storage [10]. However, with the advancement of technology and a deeper understanding of the fermentation process, fermentation technology has a wider range of applications, including the preparation of functional plant-based fermented beverages. It is now well-established that fermentation can improve food quality by enhancing flavor, nutritional value, and bioavailability [11]. At present, a variety of plant raw materials have been successfully developed into functional drinks, including tea, lychee, carrot, etc., and systematic studies have been carried out from two aspects of nutritional value and sensory properties [12,13,14]. In a recent study, Abla [15] found that the antioxidant and anti-inflammatory activities of *Rosa rugosa* “Mohong” were significantly enhanced after fermentation with *Saccharomyces rouxii*. However, the effects of fermentation on the flavor profiles of rose beverages have not been further elucidated.

Volatile organic compounds (VOCs) are critical determinants of food sensory quality and consumer acceptability [16]. Rose is a characteristic aromatic plant, containing over 400 volatile compounds including aromatic alcohols, ethers, esters, aldehydes, alkanes, oxides, monoterpenes, and sesquiterpenes [17]. Key components contributing to the characteristic rose aroma include nerol, citronellol, linalool, geraniol, nonadecane, tricosane, geranyl acetate, and eugenol [18]. Therefore, it is desirable to unravel the effect of fermentation technology on the flavor of rose beverage.

The production efficiency of flavor compounds can be affected by external factors such as microbial species, inoculation methods, and microbial interactions [19]. Studies have demonstrated that mixed-strain fermentation can offer advantages over single-strain fermentation, often enhancing the production of bioactive substances, aroma, and overall sensory quality in fermented products [20,21,22]. Thus, it was intriguing to increase flavor and special nutrients in rose beverages using multi-strain starter cultures. Yeast, particularly *Saccharomyces cerevisiae*, is known for its low production of antagonistic compounds compared to some bacteria, its antibiotic resistance, ability to neutralize enterotoxins, and its synergistic interactions with other microbes [23]. Moreover, it was shown that the abundance and diversity of aromatic compounds in fermentation products were significantly increased by the combined use of *S. cerevisiae* and *L. plantarum* [24,25]. Based on this rationale, we selected *L. plantarum* and *S. cerevisiae* to investigate their individual and combined effects on the nutritional and flavor properties of rose beverages, aiming to improve product attributes.

Headspace–solid-phase microextraction coupled with gas chromatography–mass spectrometry (HS-SPME/GC-MS) is a powerful technique for identifying and quantifying volatile compounds in food, enabling comprehensive flavor analysis [26]. Electronic sensory analysis has become the preferred method for evaluating beverage flavor profiles due to its rapidity, sensitivity, and cost-effectiveness [27]. Combining these chromatographic and electronic sensory analyses leverages their complementary strengths and is widely applied to study the flavor profiles of complex beverages [28]. 

In this research, the physicochemical indexes (such as pH, total sugar, chromatic intensity, etc.), free amino acids (FAAs), and organic acids of functional rose-based beverage fermented by *L. plantarum*, *S. cerevisiae*, and co-inoculation of *L. plantarum* and *S. cerevisiae* were characterized. The volatile profiles of the fermented beverages were subsequently analyzed using an E-nose and HS-SPME/GC-MS. This study enhances the understanding of the nutritional and flavor changes in functional rose beverages, contributes to the development of standardized and high-quality production methods, and provides valuable insight for the advancement of the rose industry.

## 2. Materials and Methods

### 2.1. Materials

Rose (*Rosa rugosa*) was grown in Kunming, Yunnan Province, China (25.08 N, 102.45 E). *L. plantarum* Picp-2 (CCTCC M20191045) was stored in the China Center for Type Culture Collection (Wuhan, China). *S. cerevisiae* SY-1 (CGMCC 2.119) was purchased from the China General Microbial Culture Preservation and Management Center (Beijing, China).

### 2.2. Sample Preparation

Cryopreserved strains were initially inoculated into MRS broth (*L. plantarum*) or YPD liquid medium (*S. cerevisiae*) and incubated at 37 °C or 30 °C, respectively, for 16–24 h to reach the logarithmic growth phase. Activated cultures were then transferred (as inoculum) to 200 mL of fresh MRS broth or YPD medium and incubated for an additional 16–24 h at 37 °C. Following centrifugation at 6000 rpm (4 °C) for 10 min, the resulting cell pellets were washed three times with an equal volume of sterile normal saline (0.9%, *w*/*v*). The activated strains were inoculated into the rose beverage substrate within 1 h to maintain cell viability.

Dried rose petals were extracted in water at 70 °C for 3 h to maximize the extraction of aromatic substances and nutrients. Sucrose (2%, *w*/*v*) was added as a supplementary carbon source. The resulting mixture was then pasteurized to obtain the unfermented rose beverage (UFR). Three types of fermented rose beverages were produced from the UFR by liquid-state fermentation, using L. plantarum alone (LFR), S. cerevisiae alone (SFR), and a co-culture of L. plantarum and S. cerevisiae (LSFR). After cooling the pasteurized beverage to 25 °C, the activated microbial inoculants were added to achieve an initial cell count of 6 log CFU/mL. Fermentation was carried out at 30 °C for 72 h. An uninoculated rose beverage was incubated under identical conditions to serve as a control.

### 2.3. Determination of Physicochemical Parameters

The pH and total sugar content were determined according to the Chinese National Standard GB/T 13662-2018 [29]. Briefly, pH was measured using an FE28 Standard pH meter (Mettler Toledo International Trade Co., Ltd., Shanghai, China). Total sugar content was quantified using the phenol-sulfuric acid method. The chromatic intensity of the rose beverages was assessed using a colorimetric method [30]. Briefly, 2 mL of each sample was placed in a cuvette, and the full visible absorption spectrum (400–700 nm) was recorded at 1 nm intervals using a UV–vis spectrophotometer (UV-vis 2450, Shimadzu Corporation, Kyoto, Japan). Prior to color analysis, samples were filtered through a 0.45 μm hydrophilic PES membrane. For trichromatic value measurements, color coordinates were determined using the CIELAB model, including L* (lightness, ranging from 0 for black to 100 for white), a* (red–green coordinate, with positive values indicating redness and negative values indicating greenness), and b* (yellow–blue coordinate, with positive values indicating yellowness and negative values indicating blueness).

### 2.4. Determination of Total Phenols and Total Flavonoids

Total phenolic content (TPC) in rose beverages was determined using the Folin–Ciocalteu assay [30]. Briefly, 1 mL of sample was mixed with 1.5 mL of Folin–Ciocalteu reagent at room temperature for 3–8 min. Subsequently, 1 mL of Na_2_CO_3_ (20%, *w*/*v*) was added, and the volume was adjusted to 10 mL with ultrapure water. After incubation for 1 h, the absorbance at 765 nm was measured. TPC was calculated based on a calibration curve prepared using gallic acid (y = 0.0063x + 0.0099, R² = 0.9982) and expressed as milligrams of gallic acid equivalents per milliliter (mg GAE/mL).

Total flavonoid content (TFC) in rose beverages was determined using the aluminum nitrate colorimetric method, with rutin as the standard [30]. Briefly, 1 mL of NaNO_2_ (5%, *w*/*v*) was added to 1 mL of the sample and gently swirled for 5 min. Approximately 1 mL of Al(NO_3_)_3_ (10%, *w*/*v*) was then added and allowed to react for 5 min, followed by the addition of 10 mL of NaOH (0.5 M). The mixture was incubated for 15 min, and the absorbance at 510 nm was measured. TFC was calculated based on a calibration curve prepared using rutin (y = 1.2954x − 0.0027, R² = 0.9996) and expressed as milligrams of rutin equivalents per milliliter (mg RE/mL).

### 2.5. Determination of Free Amino Acids (FAAs) and Organic Acids

The free amino acid composition in rose fermented beverages was quantified using an automated amino acid analyzer (LA8080, Hitachi Ltd., Kyoto, Japan) [31]. For sample preparation, 10% sulfosalicylic acid (100 μL) was added to the sample, followed by a 1-h incubation at 5 °C. The mixture was then centrifuged at 12,000 rpm for 20 min at 4 °C and filtered through a 0.45 μm hydrophilic PES membrane. A 10 μL aliquot of each prepared sample was injected onto a ZORBAX Eclipse AAA column (4.6 × 75 mm, 3.5 μm; Agilent Technologies, Waldbronn, Germany). Chromatographic separation was achieved using a mobile phase and derivatization reagent (ninhydrin) at flow rates of 0.35 mL/min and 0.3 mL/min, respectively, with a 30 μL injection volume.

Organic acid quantification (lactic, acetic, malic, and citric acids) was performed by HPLC analysis according to the Chinese National Standard GB/T 5009.157-2016 [32]. Samples were diluted 1:9 with sterile saline, homogenized by vortex mixing for 10 min, and centrifuged (12,000 rpm, 15 min). Following filtration through a 0.22 μm hydrophilic PES membrane, chromatographic separation was performed using a mobile phase consisting of 0.01 mol/L KH_2_PO_4_ buffer (pH 2.55) containing orthophosphate and methanol (97:3, *v*/*v*) at a flow rate of 0.5 mL/min. The column temperature was maintained at 30 °C. Detection was performed at 210 nm, and quantification was achieved using external standard calibration curves.

Taste characteristics were evaluated via taste activity value, calculated as the ratio of analyte concentration to its sensory detection threshold (Appendix A).

### 2.6. Determination of VOCs by HS-SPME-GC-MS

Five milliliters of rose beverage specimens were measured and transferred into 20 mL hermetic headspace vials. Using precision microinjection equipment, 6 μL of internal reference standard (Octane, 20 μg/mL) was administered. The automated AOC-6000 autosampler executed the HS-SPME protocol, wherein a specialized fiber assembly (50/30 µm DVB/CAR on PDMS Extraction Head) was positioned in the headspace compartment. The system maintained continuous agitation (250 rpm) under thermostatic control at 70 °C for 30 min. Upon completion of extraction, the fiber matrix was promptly transferred to the GC-MS injection port for thermal desorption (250 °C, 4 min), followed by subsequent chromatographic analysis.

Chromatographic separation of volatile organic compounds (VOCs) was performed using an Agilent HP-INNOWAX capillary column (60 m × 0.25 mm i.d., 0.25 μm film thickness). The injector temperature was maintained at 250 °C, and ultra-high purity helium was used as the carrier gas at a flow rate of 1 mL/min. The oven temperature program was as follows: initial hold at 40 °C for 4 min, followed by a ramp to 140 °C at 6 °C/min with a 5-min hold. Subsequent ramps were to 150 °C at 3 °C/min (1 min hold), 197 °C at 5 °C/min (1 min hold), 205 °C at 1 °C/min, and finally to 240 °C at 7 °C/min with a 10-min hold.

Mass spectrometric detection was performed using electron ionization (EI) at 70 eV. The ion source and detector temperatures were set at 230 °C, and the quadrupole temperature was maintained at 150 °C. Full-scan acquisition was performed from *m*/*z* 40 to 400 with a dwell time of 250 ms per mass channel.

### 2.7. Qualitative and Quantitative Analysis

Chromatographic peak deconvolution and analysis were performed using the AMDIS algorithm integrated into TraceFinder 5.1 software. Compound identification was achieved through spectral matching against the NIST reference database, with additional validation via retention index (RI) comparisons. The RI values were determined through linear interpolation of n-alkane standards (C_7_–C_40_) using the formula: RI = 100n + 100 × (t_a_ − t_n_)/(t_n+1_ − t_n_), where n represents the carbon count of the alkane homolog, t_a_ corresponds to analyte retention time, while t_n_ and t_n+1_ denote retention times of consecutive n-alkanes. Quantitative analysis was conducted using an internal standard semi-quantitative method with the equation: C_i_ = (C_is_ × A_i_)/A_is_, where C_i_ indicates internal standard concentration, and A_i_/A_is_ represents the analyte-to-standard peak area ratio [33].

The relative odor activity value (ROAV) was used to identify key aroma compounds contributing to the overall flavor profile of the samples. ROAV is calculated as the ratio of the compound’s concentration to its sensory detection threshold, reflecting its potential contribution to the aroma. Compounds with ROAV ≥1 were considered key volatile compounds. The calculation formula is as follows: ROAV_i_ = C_i_/T_i_. Among them, ROAV_i_ is the relative odor activity value of compound i, and C_i_ is the relative content of the compound; T_i_ is the threshold of the compound [31].

### 2.8. Analysis of E-Nose

The odor property of the rose beverage was analyzed by a PEN 3 E-nose system (Airsense Analytics Co. Ltd., Schwerin, Germany) [30]. The test conditions were set for 200 s of sensor cleaning, automatic zero return time of 5 s, pre-sampling time of 5 s, detection time of 120 s, gas injection rate, and carrier gas speed of 200 mL/min.

### 2.9. Data Analysis

All experiments, including fermentation, were performed in at least triplicate independent replicates. Data are presented as the mean ± standard deviation (SD). Data visualization was performed using GraphPad Prism 8.0.2. Statistical significance was evaluated using one-way analysis of variance (ANOVA) followed by the least significant difference (LSD) test. Differences among group means were assessed using Duncan’s multiple range test at a 95% confidence level (*p* < 0.05) using SPSS Statistics 19.0 software (IBM SPSS, Chicago, IL, USA). Principal component analysis (PCA) and partial least-squares discriminant analysis (PLS-DA) were conducted using the web-based platform MetaboAnalyst 5.0.

## 3. Results and Discussion

### 3.1. Physicochemical Characteristics Analysis

The physicochemical parameters of the six basic components of rose-based functional beverage fermented by different strain groups were analyzed (Table 1). Microbial growth during fermentation necessitates the consumption of sugars for conversion into other metabolites. In all three fermentation groups, the inoculated microorganisms exhibited robust growth, maintaining cell counts above 7.37–8.46 log CFU/mL after 24 h, indicating that the rose beverage substrate was conducive to the growth of both lactic acid bacteria and yeast. The total reducing sugar content in the unfermented control group was 36.52 g/L. Following fermentation, a significant decrease in total reducing sugar was observed, with reductions of 63.39% in the SFR group, 79.46% in the LFR group, and 88.80% in the LSFR group. This reduction in sugar content correlated with the observed changes in pH. After 3 days of fermentation, the pH value of the rose beverages significantly decreased, which is primarily attributed to the production of organic acids by the lactic acid bacteria strains.

Following fermentation, the free amino acid (FAA) content in the rose beverage increased significantly, by 6.75-fold in LFR samples and 4.96-fold in LSFR samples. This observation aligns with previous studies, which reported that increased amino acid content during fermentation is associated with the decomposition of proteins and peptides by microbial peptidases and autolysis of lactic acid bacteria [34,35]. In contrast, the total amino acid content in the rose beverage fermented solely by *S. cerevisiae* (SFR) was significantly reduced. Amino acids are known to serve as important precursors for the formation of volatile compounds that contribute to the aroma of beverages during yeast metabolism [36]. Therefore, it is plausible that the conversion of amino acids into alcohols, aldehydes, ketones, and other flavor compounds contributed significantly to this observed reduction in the SFR group.

Fermentation by different microorganisms can lead to significant variations in the color of rose beverages. We therefore analyzed the chromatic intensity and trichromatic values of the fermented rose beverages. The results showed that the chromatic intensity was lowest in the SFR group (0.87) and highest in the LFR group (1.02). This suggests that the higher chromatic intensity in the LFR group may be related to the formation of polymer pigments following *L.* plantarum fermentation, potentially influenced by the decrease in pH. The L* values (lightness) of the fermented rose beverages followed the order: SFR > LSFR = UFR > LFR. The a* values (redness) showed the order: LFR > LSFR > SFR > UFR, while the b* values (yellowness) were in the order: UFR > SFR > LFR > LSFR. These results indicate that fermentation induced a shift in the color of the rose beverages, moving towards bluer and redder hues. Previous research has demonstrated that beverage color is influenced not only by the raw materials but also by reactions such as the Strecker and Maillard reactions that occur during fermentation [30]. The observed color differences among beverages fermented with different strains are likely correlated with the extent and nature of these reactions. 

Total phenolic content (TPC) and total flavonoid content (TFC) are crucial secondary metabolites in food products. In this study, both TPC and TFC in the rose beverages significantly increased after fermentation across all treatment groups compared to the unfermented control. Specifically, TPC and TFC increased by 11.26% and 15.00% in SFR, by 18.92% and 25.00% in LFR, and by 23.87% and 25.00% in LSFR. These findings are consistent with those of Tlais et al. [37], who reported that fermentation by both L. plantarum and *S. cerevisiae* significantly (*p* < 0.05) increased TPC and TFC in apple by-products, with the most substantial increases observed in co-cultured samples (*p* < 0.05). This further supports the notion of synergistic effects between *S. cerevisiae* and *L. plantarum* on metabolic activities. Furthermore, the increase in total phenols and flavonoids during fermentation is related to the hydrolase production capacity of the fermentation strains and the biotransformation of these compounds, which are influenced by the metabolism of sugars, organic acids, and amino acids [38]. A comparative study by Jin et al. [39] on fermented mango juice highlighted that the transformation ability of L. plantarum regarding TPC was significantly stronger than that of *S. cerevisiae*.

### 3.2. Free Amino Acids and Organic Acids Analysis

The concentrations of free amino acids (FAAs) and organic acids in the fermented functional rose beverages were also analyzed (Figure 1). FAAs play a significant role in the aroma and flavor of food and serve as precursors for many volatile compounds [40]. FAAs can be broadly categorized based on their taste profiles: sweet (Gly, Ser, Ala, Pro, Thr, Cys, Met), bitter (Arg, His, Ile, Leu, Phe, Lys, Tyr, Trp, Val), and umami (Asp, Glu, Asn) [41]. The relative proportions of these taste categories within the total FAAs were as follows: UFR (23.17% sweet, 17.43% bitter, 59.40% umami), LFR (45.74% sweet, 31.84% bitter, 22.41% umami), SFR (26.10% sweet, 21.31% bitter, 52.59% umami), and LSFR (43.71% sweet, 31.25% bitter, 25.03% umami). These results indicate that fermentation by *L. plantarum* significantly contributed to the conversion of umami amino acids into sweet and bitter amino acids, and the metabolic impact of L. plantarum on the amino acid distribution in the functional rose beverage was considerably greater than that of S. cerevisiae. Furthermore, following fermentation by lactic acid bacteria (LFR and LSFR), the total aromatic amino acid content increased, particularly tryptophan (Trp) and phenylalanine (Phe), which showed substantial increases of 28.73-fold and 20.31-fold, respectively, in LFR, and 5.09-fold and 3.86-fold in LSFR. Previous studies suggest that the degradation of Trp by microorganisms can promote the formation of aromatic compounds [42], while the increase in phenylalanine is conducive to the formation of volatile compounds such as benzyl alcohol, phenylethanol, and benzene derivatives [43], thereby enriching the flavor profile of rose beverages. Additionally, in the LFR and LSFR groups, the content of essential amino acids (Val, Leu, Arg, Trp) significantly increased by 9 to 28 times, substantially enhancing the nutritional value of these fermented beverages [44]. The taste activity values (TAVs) of FAAs were calculated to assess their contribution to the flavor of rose beverages (Figure 1B). However, none of the free amino acids exhibited a TAV greater than 1, suggesting they did not have a significant direct impact on the overall taste perception in this study.

As depicted in Figure 1C, notable differences were observed in the types and concentrations of organic acids across the four sample groups. Lactic acid, acetic acid, and citric acid were undetectable in the UFR sample, while lactic acid was absent in the SFR sample. Following fermentation, the total organic acid content increased, with the highest concentrations observed in LFR (8.23 g/L), followed by LSFR (6.29 g/L), and then SFR (2.18 g/L). Similar trends have been reported in tea infusions fermented with L. plantarum and Saccharomyces boulardii [45]. Malic acid, tartaric acid, and succinic acid were identified as the primary organic acids in the unfermented rose beverage, consistent with the findings of Önder et al. [46]. Lactic acid is a characteristic metabolic product of lactic acid bacteria fermentation [47]. Significant increases in lactic acid content were observed in the LFR and LSFR groups (*p* < 0.01) (Figure 1B). At the end of fermentation, lactic acid accounted for 0%, 57.74%, 52.35%, and 0% of the total organic acids in UFR, LFR, LSFR, and SFR, respectively, indicating its major role in determining the acidity differences among the fermented beverages. Notably, the lactic acid content in LSFR was significantly lower than in LFR. Previous research suggests that certain yeasts can utilize lactic acid as a carbon source, which may explain the reduced lactic acid concentration observed in the co-fermented sample (LSFR) [48]. Acetic acid, a short-chain fatty acid, is recognized for its importance in regulating intestinal health and its potential impact on inflammatory bowel diseases (IBD) [49]. While undetectable in the unfermented rose beverage, significant accumulation of acetic acid was observed in the fermented samples, particularly those fermented with *L. plantarum*. This accumulation undoubtedly contributes to the enhanced nutritional properties of the fermented rose beverage.

The TAVs of organic acids were calculated to evaluate their contribution to the flavor of rose beverages (Figure 1D). With the exception of citric acid and lactic acid in the SFR group (TAVs < 1), six organic acids exhibited TAVs greater than 1 in the fermented rose beverages. This suggests that these organic acids with TAVs > 1 likely contribute significantly to both the flavor and taste perception of the fermented rose beverages. Malic acid, lactic acid, acetic acid, and citric acid impart a fresh, acidic taste [50]. Succinic acid can contribute umami, sourness, and astringency. The synergistic interactions among these organic acids can enhance the body and modulate the sweetness of the rose beverage, resulting in a pleasant sweet-and-sour profile.

### 3.3. Electronic Nasal Analysis

Flavor is a critical characteristic influencing beverage quality and consumer acceptance. Electronic sensory analysis, such as the electronic nose (E-nose), is widely employed for objective and rapid flavor profiling, offering advantages over traditional sensory evaluation methods due to its simplicity, speed, and objectivity [51]. We utilized E-nose analysis to investigate the olfactory characteristics of rose beverages fermented with different microbial combinations. As shown in Figure 2A, the response patterns of the SFR and LSFR groups (associated with *S.* cerevisiae fermentation) were similar but exhibited slight differences in intensity across several sensors, including W1S (sensitive to methyl groups), W2S (sensitive to alcohols, aldehydes, and ketones), W1W (sensitive to sulfides), W2W (sensitive to aromatic compounds and organic sulfides), W1C (sensitive to aromatics), and W5S (sensitive to nitrogen oxides). In contrast, the LFR group displayed higher response intensities for W1W and W2W but lower intensities for W1S, W2S, W1C, and W5S, distinguishing its profile from those of SFR and LSFR.

Principal component analysis (PCA) was applied to the E-nose response signals to visualize the differences in volatile profiles among the rose beverages fermented with different strains. As shown in Figure 2B, the first two principal components (PC1 and PC2) accounted for 99.4% of the total variance, effectively capturing the key information regarding the volatile compounds. The unfermented samples clustered closely with the samples fermented solely by L. plantarum (LFR), suggesting that lactic acid bacteria fermentation largely preserves the inherent flavor characteristics of the rose beverage. Conversely, the three fermented rose beverage samples (LFR, SFR, LSFR) were distinctly separated in the PCA plot (Figure 2B), indicating that each fermentation condition resulted in a unique flavor profile. This observation is consistent with the distinct patterns observed in the radar plot (Figure 2A). Therefore, the E-nose analysis successfully discriminated the flavor characteristics of the three fermented rose beverages to a certain extent. However, a more in-depth and effective understanding of the influence of different inoculation methods on flavor profiles necessitates the integration of additional analytical techniques.

### 3.4. Volatile Compounds Analyzed by HS-SPME-GC-MS

The volatile profiles of the four rose beverage samples (UFR, LFR, SFR, and LSFR) were analyzed using HS-SPME coupled with GC-MS. A total of 245 volatile compounds were identified and classified into various chemical groups: 52 alcohols, 27 esters, 19 ketones, 21 aldehydes, 8 phenols, 15 benzene compounds, 7 acids, 51 terpenes, 18 alkanes, 8 ethers, and 19 other compounds. Specifically, 202 compounds were identified in UFR, 223 in LFR, 181 in SFR, and 222 in LSFR. These results demonstrate that fermentation, particularly by lactic acid bacteria, enriched the diversity of volatile substances in the rose beverages, aligning with findings from previous studies [22,52].

A Venn diagram was constructed based on the volatile compounds identified in Appendix A (Figure 3A) to visualize the relationships between the four rose beverage samples. The observed differences in volatile compound profiles among the groups are directly attributable to the distinct metabolic activities of the different microbial fermentations. LFR and LSFR shared 214 compounds, while SFR and LSFR shared 164 compounds, indicating a greater similarity between the aroma profile of LSFR and LFR. Each beverage type also contained unique volatile compounds. Only one unique volatile compound was identified in the UFR sample, suggesting that most of the original volatile compounds were altered during fermentation by the single or co-cultures, potentially serving as substrates for biotransformation. In total, 1, 10, and 2 unique compounds were found exclusively in the LFR, SFR, and LSFR samples, respectively, likely arising from the specific metabolic pathways active in each microbial treatment.

Principal component analysis (PCA), a multivariate statistical technique, was applied to the quantitative data of the 245 volatile compounds to highlight the differences among the samples [53]. The PCA score plot (Figure 3B) effectively separated the four rose beverage samples based on the first two principal components (PC1 and PC2), which accounted for 76.3% and 17.7% of the total variance, respectively, with a cumulative contribution of 94%. The separation between the sample groups was more pronounced than the variation within each group, indicating that the different microbial fermentations had a significant impact on the volatile metabolome of the functional rose beverages. The unfermented rose beverage (UFR) showed the largest deviation from the other samples, highlighting the substantial changes in volatile compounds induced by fermentation. The greater overlap between the LSFR and LFR groups compared to the overlap between LSFR and SFR suggests that *L. plantarum* exerted a more dominant influence on the volatile flavor substances compared to *S. cerevisiae* in the co-fermentation.

### 3.5. Classification Comparison of Volatile Compounds in Rose Beverage

A comparative analysis was conducted on the heatmap of volatile organic compounds in rose beverages after fermentation. The heatmap provides a clear visualization of the volatile components. Significant variations in alcohols, esters, aldehydes, ketones, acids, phenols, and terpenes were identified, likely contributing to the flavor differences. In all samples, alcohols were the dominant class of flavor compounds, which was an important contributor to the fragrance of rose, and was also the highest content of aromatic compounds [54]. Despite the involvement of S. cerevisiae in the SFR and LSFR fermentations, ethanol was not detected. This absence may be attributed to its conversion into ethyl esters by the microorganisms [55]. Following fermentation, the content of inherent alcohols such as citronellol (3,7-dimethyl-6-octen-1-ol), linalool (3,7-dimethylocta-1,6-dien-3-ol), and geraniol ((E)-3,7-dimethylocta-2,6-dien-1-ol) decreased in the *S. cerevisiae* monoculture sample (SFR), while their types and content significantly increased in the samples associated with LAB fermentation (LFR and LSFR). This suggests that the presence of LAB is conducive to enhancing the alcohol-derived flavor characteristics of the rose beverage. Furthermore, LAB fermentation can also promote the conversion of certain low-abundance alcohols, including benzyl alcohol, 1-nonanol (nonan-1-ol), 2-undecanol (undecan-2-ol), 2-nonanol (nonan-2-ol), and 1-hexanol (hexan-1-ol). Benzyl alcohol, 1-nonanol, and 1-hexanol have been identified as critical volatile organic compounds influencing the flavor profiles of various fermented products, such as Pu’er tea [56], fermented oat beverage [57], and greengage wine [58]. The increase in these flavor compounds may be due to the hydrolysis activity of β-glucosidase produced by LAB [59]. This accumulation could contribute a richer aroma profile, including floral, fruity, and sweet notes, to the fermented rose beverages. Interestingly, the content of phenylethyl alcohol increased significantly in all fermented samples, with increases of 10.06-fold in LFR, 5.12-fold in LSFR, and 5.63-fold in SFR compared to the UFR. Phenylethyl alcohol is a key floral aroma compound and one of the most abundant aromatic alcohols in rose [60]. It is synthesized from L-phenylalanine via aromatic amino acid decarboxylase or aminotransferase, followed by reduction by phenylealdehyde reductase. Despite the presence of LAB, the increase in phenylethyl alcohol content in the LSFR sample was lower than in LFR, which might be a result of synergistic interactions influencing aromatic compound generation during the co-fermentation process.

Esters, typically formed through the esterification of short-chain acids with alcohols, contribute floral and fruity notes to rose beverages and can mitigate the harshness of fatty acids and the bitterness of amino acids [61]. In the present study, most ester compounds in the LAB monoculture sample (LFR) did not show significant changes, with the exception of 2-phenylethyl acetate, methyl salicylate (methyl 2-hydroxybenzoat), pentanoic acid, (3,7-dimethyl-6-octenyl) 4-methylpentanoate, and 2-isopropenyl-5-methyl-4-hexen-1-yl acetate. Methyl salicylate, characterized by a wintergreen aroma, can be released from glucoside precursors through glucosidase activity [62]. Several studies have reported that L. plantarum can produce glucosidase [63], suggesting that the increased content of methyl salicylate in LFR samples is closely linked to this enzymatic hydrolysis. Conversely, approximately half of the ester compounds were significantly upregulated in the SFR sample, indicating that yeast fermentation strongly promoted the conversion of acids and alcohols into esters. This observation is consistent with the decreasing trend in alcohol content observed in the same sample. Among the esters, isoamyl acetate (3-methylbutyl acetate) and ethyl octanoate were the most significantly upregulated in SFR samples, imparting a stronger fruity aroma to the rose beverages. Furthermore, we observed that the content of 2-phenethyl acetate was higher in the co-fermented rose beverage (LSFR) than in the monoculture samples. 2-phenylethyl acetate is an aromatic ester found in fermented rose beverages, possessing floral, fruity, and honey-like aromas. It can be formed by the esterification of 2-phenylethyl alcohol with acetate or through the transesterification with acetate [64]. Our earlier analysis showed that the phenylethyl alcohol content was lowest in the LSFR sample among the fermented beverages, providing a plausible explanation for the origin of 2-phenethyl acetate. This result suggests that co-culturing L. plantarum with S. cerevisiae could promote the conversion of 2-Phenethyl acetate from phenylethyl alcohol.

Terpenes (monoterpenes and sesquiterpenes) are considered major aromatic compounds in roses. A total of 51 terpenes were detected across the rose samples (Appendix A). Notably, four novel volatile compounds were identified in the fermented rose beverages, potentially due to the synthesis of specific terpenes by the fermenting strains. Among the detected terpenes, 35 compounds were upregulated in the LFR group, while only 9 compounds were upregulated in the SFR group. This finding differs from a study by Quan et al. [65], which reported reduced terpene content in orange juice after LAB fermentation. This discrepancy might be attributed to strain-specific characteristics. The *L. plantarum* strain used in this study has strong β-glucosidase production and can release terpenoids through hydrolysis to enrich the aroma of rose beverages [66].

Aldehydes and ketones contribute to the complexity of food flavor, modulate aroma release, and play a decisive role due to their low odor thresholds [67]. Following fermentation, the content of most aldehydes, including benzaldehyde, furfural, and nonanal, decreased, consistent with previous findings [21]. As flavor components in food, aldehydes were unstable and could easily be converted into acids and alcohols under fermentation, which might be the major reason for their decreased levels [68]. 2,4-Dimethyl benzaldehyde exhibited the highest content in aldehydes, which was increased by 4.6- and 3.8-fold in LFR and LSFR after fermentation. A previous study has shown that 2,4-dimethyl benzaldehyde was likely the primary contributor to the sour odor produced by LAB [69]. Hence, the addition of yeast can effectively improve the sour flavor brought by LAB fermentation. Ketone is a predominant volatile compound in many fermented foods, especially in yogurt [70]. In this study, we identified 19 ketone compounds in four rose beverages, of which 1 and 3 ketones were significantly upregulated in SFR and LFR samples, respectively. In addition, we also find that the relative contents of three flavor compounds, including isophorone (3,5,5-trimethylcyclohex-2-en-1-one) [71], 2-heptanone (heptan-2-one) [21], and 2-nonanone (nonan-2-one) [72], were much higher in the co-fermentation sample than those in the monocultured samples. This result further confirmed that the synergistic effect of *L. plantarum* and *S. cerevisiae* is conducive to the accumulation of flavor substances.

Acids enhance food flavor by staying below the odor threshold and balancing unpleasant odors [73]. In the rose beverages, seven aromatic active acid compounds were detected: geranic acid ((E)-3,7-dimethyl-2,6-octadienoic acid), decanoic acid, octanoic acid, citronellic acid (3,7-dimethyl-6-octenoic acid), nonanoic acid, hexanoic acid, and 2-methylbutyric acid. Notably, SFR and LSFR samples had relatively high levels of decanoic acid and octanoic acid, while LFR and LSFR samples had higher levels of nonanoic acid and citronellonic acid. This suggests that L. plantarum may enhance long-chain fatty acid formation, while *S. cerevisiae* may promote short-chain fatty acid formation in rose beverages. Lu et al. [74] and Li et al. [21] also reported that yeast and LAB fermentation increased the content of fatty acids, respectively.

Phenylpropanoids and benzoyl compounds originate from the aromatic amino acid phenylalanine and were significant contributors to the aroma of plant-based foods [75]. The most dominant volatiles of the phenolics in rose beverage were eugenol (4-allyl-2-methoxyphenol) (strong aroma of cloves), methyleugenol (4-allyl-1,2-dimethoxybenzene), and 2,4-di-tert-butylphenol (2,4-bis(1,1-dimethylethyl)phenol), which were important aroma compounds in rose plants (Figure 4) [76]. After fermentation, its content decreased by 69.1–93.6% in *S. cerevisiae* monoculture rose beverage, while it could maintain a high relative content in the *L. plantarum* associated samples, especially for eugenol and 2,4-ditert-butylphenol. The content of them under LAB treatment increased by 6.16- and 3.74-fold, and 2.69- and 1.69-fold in LFR and LSFR, respectively, compared with unfermented rose beverage. Previous studies showed that many LAB produce β-glucosidase, and the increase in eugenol content in LFR and LSFR may be closely related to β-glucosidase hydrolysis [77].

Generally, environmental variables, such as temperature, dissolved oxygen, and pH, are crucial in controlling microbial metabolic pathways. However, the metabolic products of microorganisms play a decisive role in the formation of flavor [78,79]. Therefore, in the process of exploring the flavor formation mechanism of fermented rose beverages, the influence of environmental variables should also be considered. In future research, we will further explore the influence of environmental variables on the flavor of fermented rose beverages, thereby comprehensively analyzing the rules by which different strains regulate the formation of characteristic flavors.

### 3.6. ROAV Analysis

A total of 245 volatile compounds were identified by gas chromatography–mass spectrometry (GC-MS), and their ROAVs were determined. As shown in Table 2, a total of 34 aromatic compounds with ROAV ≥1 were identified across the four rose beverage samples. Among them, the UFR, SFR, LFR, and LSFR samples contained 27, 25, 29, and 32 compounds, respectively. Based on the ROAV values and the classification by Gu et al. [80], the aromas were categorized into three groups: light (1 ≤ ROAV < 10), medium (10 ≤ ROAV < 100), and intensive (ROAV ≥ 100). Intensive aromas included eugenol, D-citronellol, linalool, citronellol, geraniol, phenylethyl alcohol, β-myrcene (7-methyl-3-methyleneocta-1,6-diene), 1-nonanol, 2,4-di-tert-butylphenol, 2-methoxy-2-methylbutane, isoamyl acetate, D-limonene ((4R)-4-isopropenyl-1-methylcyclohexene), nonanal, 2-phenylethyl acetate, ethyl hexanoate, 2-(methylamino)benzoic acid methyl ester, β-phellandrene (1-methyl-4-(1-methylethenyl)cyclohexene), β-nerol ((2Z)-3,7-dimethylocta-2,6-dien-1-ol), 2-undecanol. Compounds such as methyl salicylate, α-Terpineo, methyleugenol, (Z)-3,7-Dimethyl-2,6-octadienylethanoate, α-phellandrene ((5S)-5-isopropyl-2-methylcyclohexa-1,3-diene), 2-nonanol, 1-decanol (decan-1-ol), 1-hexanol, benzaldehyde, and ethyl octanoate had medium aromas.

A heatmap (Figure 5) was created to clearly show the contribution of volatile compounds with ROAV ≥1 in different rose beverages. In the LFR, the aroma contribution of compounds such as methyl salicylate (apple aroma), linalool (rose aroma), nonanal (floral aroma), citronellol (rose and fruity aroma), geraniol (rose aroma), and eugenol (cloves aroma) was significant. In the LSFR, the aroma contribution of compounds like 1-decanol (rose aroma), 2-phenylethyl acetate (rose and fruity aroma), β-nerol (rose and orange aroma) was notable. In the SFR, the aroma contribution of compounds including ethyl octanoate (pineapple aroma), ethyl hexanoate (fruity aroma), and isoamyl acetate (banana aroma) was evident. However, in the UFR, the aroma contribution of compounds such as (E)-β-ionone (sweet citrus aroma), benzaldehyde (nut aroma), and nonanal (rose and orange aroma) were pronounced.

### 3.7. Identification of Flavor Markers Using PLS-DA

The differences in volatile compounds among different fermented rose beverages were revealed through multivariate statistical analysis performed using PLS-DA. The findings revealed that the model did not exhibit overfitting; the PLS1 and PLS2 explained 65.6% and 28.1% of the total variance, respectively, which can account for the differences in volatile compounds (Figure 6A). The potential flavor markers in rose beverages can be identified using the VIP score, which reflects the influence of volatile compounds on the overall flavor profile. When VIP > 1, these markers were likely to significantly influence the flavor of the samples. In Figure 6B, compounds such as eugenol, 2-methoxy-6-(2-propenyl)phenol, phenylethyl alcohol, (E)-3,7-dimethyl-2,6-octadienoic acid, methyleugenol, (2r,2α,4aα,8aβ)-1,2,3,4,4a,5,6,7-octahydro-α,α,4a,8-tetramethylnaphthalen-2-ylmethanol, ethyl octanoate, citronellol, 2,4-dimethylbenzaldehyde, [2r-(2α,4aα,8aβ)]-decahydro-α,α,4a-trimethyl-8-methylenenaphthalen-2-ylmethanol, D-citronellol, (3,7-dimethyl-6-octenyl)4-methylpentanoatel, 2,4-di-tert-butylphenol exhibited VIP > 1. Among these, eugenol, phenylethyl alcohol, (E)-3,7-dimethyl-2,6-octadienoic acid, methyleugenol, ethyl octanoate, citronellol, D-citronellol, and 2,4-di-tert-butylphenol have OAV > 1, indicating that they can be identified as biomarkers and have important roles in influencing the flavor of fermented rose beverages. However, the aroma of food is formed through the interactions of volatile compounds [16]. The flavor mechanisms of key volatile compounds in rose beverages will be further investigated in future research.

The escalating consumer demands for premium beverages, driven by improved living standards, have accelerated product diversification in the beverage industry. While plant-based alternatives face challenges including flavor monotony, nutritional limitations, textural deficiencies, and stability concerns compared to dairy products, microbial fermentation has emerged as an effective strategy for enhancing both organoleptic properties and nutritional profiles of plant-derived beverages [81]. At present, it has become a recognized fact that the fermentation of beneficial active bacteria can effectively improve the flavor of plant-based beverages and enhance their nutritional value [81]. For a long time, the processing technology of edible roses has been single, and the economic benefits are not high, which has restricted the development of its industry. In this study, we prepared a fermented rose beverage for the first time, providing a new idea for the diversified development of roses. We also confirmed that the co-fermented rose beverage of *L. plantarum* and *S. cerevisiae* can not only enhance the nutritional value, but also endow the product with a special flavor. In addition, a preliminary understanding of the composition of key flavor compounds was obtained through mass spectrometry analysis. Ultimately, this knowledge will lead to more efficient optimization of the fermentation process and improved flavor and quality of functional rose beverage. Future investigations should focus on elucidating the underlying mechanisms of aroma modulation by dominant microbiota, probiotic bioactivity enhancement pathways, and systematically investigating microbial community dynamics under varied environmental parameters. These research directions will provide critical insights for valorizing rose through bioprocessing technologies and advancing sustainable development in functional beverage innovation.

## 4. Conclusions

In conclusion, our data demonstrate that co-fermentation with *Lactiplantibacillus* plantarum and *Saccharomyces* cerevisiae significantly improves the nutritional profile of rose beverage by increasing the diversity and abundance of organic acids and amino acids, as well as the total polyphenol and flavonoid content. Furthermore, the impact of liquid-state fermentation with different microbial strains on the aroma characteristics of rose beverages was comprehensively investigated using HS-SPME/GC-MS and E-nose analyses. The volatile compound profiles and aroma attributes of the rose beverage samples exhibited significant variations depending on the specific microorganisms used for fermentation. Notably, the co-fermented beverage (LSFR) combined the flavor characteristics observed in the single-strain fermentations (LFR and SFR) and developed a unique aroma profile, presenting an overall richer and more complex aroma compared to LFR and SFR alone. Using HS-SPME/GC-MS analysis, a total of 34 volatile compounds with relative odor activity values (ROAV) ≥1 were identified as contributors to the aroma. Through partial least-squares discriminant analysis (PLS-DA) modeling coupled with one-way ANOVA, eight volatile compounds were identified as significant markers impacting the aroma of the fermented rose beverages: eugenol, phenylethyl alcohol, (E)-3,7-dimethyl-2,6-octadienoic acid, methyleugenol, ethyl octanoate, citronellol, D-citronellol, and 2,4-di-tert-butylphenol. These compounds collectively contributed fruity, clove, rose, and sweet notes to the rose beverage aroma. This study enhances our understanding of the aroma composition and its modulation by various microorganisms in fermented rose beverages, providing insights for refining rose-based products. Future research will focus on elucidating the interactions among these key compounds to further uncover the underlying mechanisms of flavor formation and development.

## Figures and Tables

**Figure 1 foods-14-01868-f001:**
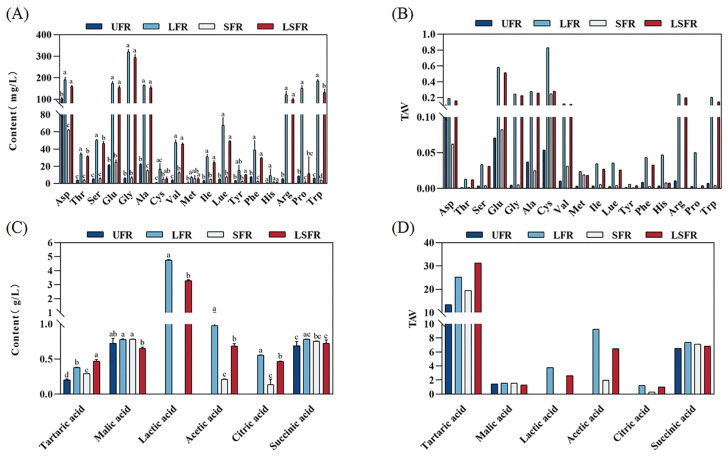
Analysis of free amino acids and organic acids of the rose beverages. (**A**) Free amino acid concentrations. (**B**) Taste activity values (TAVs) of free amino acids. (**C**) Organic acids concentrations. (**D**) TAVs of organic acids. TAVs of free amino acids and organic acids were calculated by the ratio of the concentration of a compound to its taste threshold. Values marked with different letters in the same color column of figures (**A**,**C**) were significantly different (*p* < 0.05, *n* = 3).

**Figure 2 foods-14-01868-f002:**
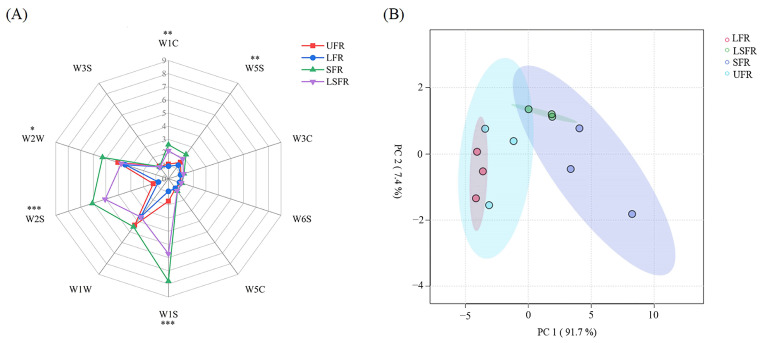
Discriminant analysis of rose beverages fermented with different microorganisms based on the e−nose data (*n* = 3). (**A**) Radar plot of E−nose sensor response, significance: *** *p* < 0.001; ** *p* < 0.01; * *p* < 0.05. (**B**) Principal component analysis plot of E−nose results for fermented rose beverage samples, the contribution rates of PC1 and PC2 are 91.7% and 7.4%. UFR: rose beverage without fermentation; LFR: rose beverage fermentation by *L. plantarum*; SFR: rose beverage fermentation by *S. cerevisiae*; LSFR: rose beverage fermentation by *L. plantarum* and *S. cerevisiae* collectively.

**Figure 3 foods-14-01868-f003:**
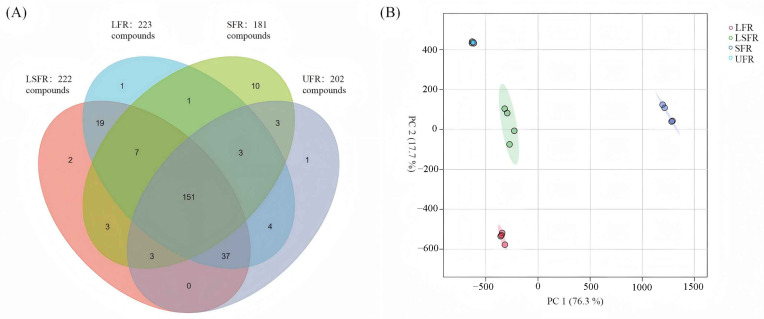
(**A**) Venn diagram. The numbers represent the quantities of volatile compounds; (**B**) Principal component analysis (PCA) of 245 volatile compounds in four rose beverage samples (*n* = 4). UFR: rose beverage without fermentation; LFR: rose beverage fermentation by *L. plantarum*; SFR: rose beverage fermentation by *S. cerevisiae*; LSFR: rose beverage fermentation by *L. plantarum* and *S. cerevisiae* collectively.

**Figure 4 foods-14-01868-f004:**
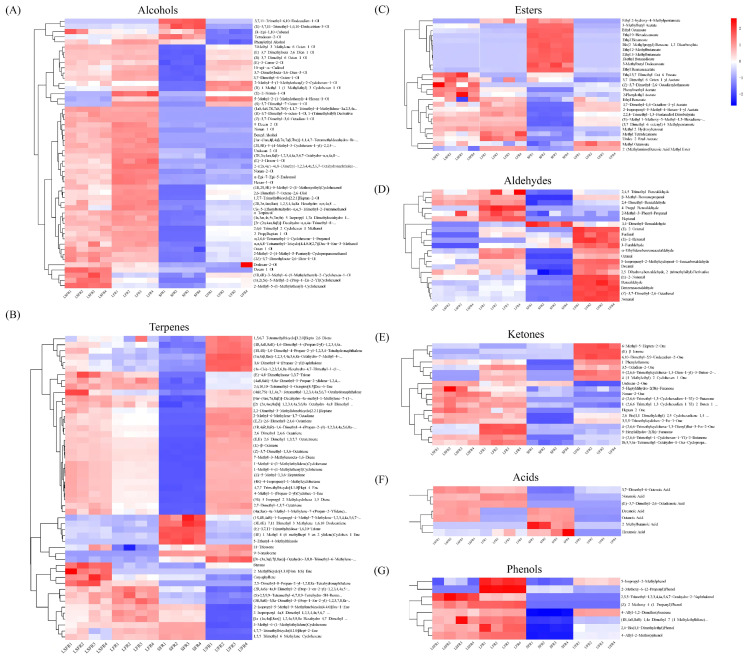
Heatmap and cluster analysis of aroma compounds in four rose beverage samples (*n* = 4). (**A**) Alcohol compounds; (**B**) Terpene compounds; (**C**) Ester compounds; (**D**) Aldehyde compounds; (**E**) Ketone compounds; (**F**) Acid compounds; (**G**) Phenol compounds. UFR: rose beverage without fermentation; LFR: rose beverage fermentation by *L. plantarum*; SFR: rose beverage fermentation by *S. cerevisiae*; LSFR: rose beverage fermentation by *L. plantarum* and *S. cerevisiae* collectively.

**Figure 5 foods-14-01868-f005:**
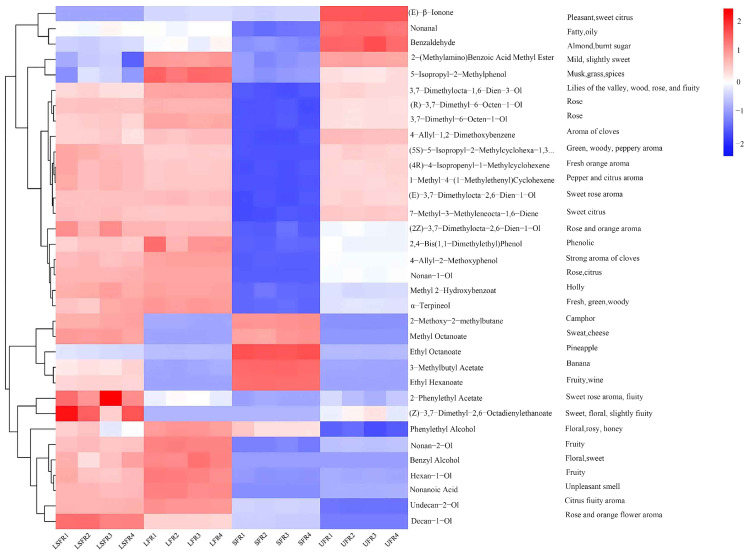
Heatmap and cluster analysis of major aroma compounds in rose beverage samples fermented with different strains. UFR: rose beverage without fermentation; LFR: rose beverage fermentation by *L. plantarum*; SFR: rose beverage fermentation by *S. cerevisiae*; LSFR: rose beverage fermentation by *L. plantarum* and *S. cerevisiae* collectively.

**Figure 6 foods-14-01868-f006:**
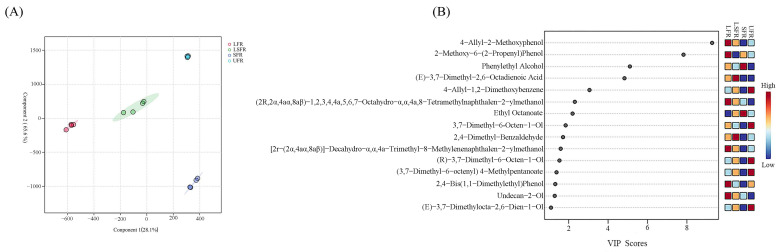
PLS−DA was conducted on data obtained from HS−SPME/GC−MS. PLS-DA score plots (**A**) and VIP score plots (**B**) for HS−SPME/GC−MS data.

**Table 1 foods-14-01868-t001:** Physical and chemical indexes related to rose fermented beverages by different bacteria.

Indexes	LFR	SFR	LSFR	UFR
pH	3.31 ± 0.006 ^d^	4.41 ± 0.020 ^c^	3.56 ± 0.006 ^b^	4.89 ± 0.006 ^a^
Total sugar (g/L)	7.5 ± 0.301 ^c^	13.37 ± 1.367 ^b^	4.09 ± 0.784 ^d^	36.52 ± 0.410 ^a^
Viable counts (log CFU/mL, 0 h) Viable counts (log CFU/mL, 24 h)	6.48 ± 0.010 ^a^ 8.46 ± 0.012 ^a^	6.79 ± 0.018 ^b^ 7.37 ± 0.085 ^b^	6.52 ± 0.023 ^a^ 8.46 ± 0.053 ^a^	0 ^c^ 0 ^c^
Total amino acid content (mg/L)	1628.38 ± 23.810 ^d^	165.27 ± 4.193 ^c^	1255.23 ± 22.288 ^b^	209.97 ± 9.053 ^a^
Chromaticity strength	1.02 ± 0.022 ^a^	0.87 ± 0.009 ^c^	0.91 ± 0.004 ^b^	0.98 ± 0.003 ^a^
L*	57.10 ± 0.590 ^c^	62.51 ± 0.076 ^b^	61.57 ± 0.368 ^a^	61.53 ± 0.284 ^a^
a*	53.31 ± 0.856 ^d^	30.22 ± 0.012 ^c^	44.19 ± 0.387 ^b^	27.51 ± 0.234 ^a^
b*	28.74 ± 0.236 ^d^	38.85 ± 0.480 ^c^	27.99 ± 0.176 ^b^	45.39 ± 0.320 ^a^
Total phenols (mg/mL)	2.64 ± 0.050 ^ab^	2.47 ± 0.020 ^b^	2.75 ± 0.020 ^a^	2.22 ± 0.06 0 ^c^
Total flavonoids (mg/mL)	0.25 ± 0.010 ^a^	0.23 ± 0.006 ^a^	0.25 ± 0.004 ^a^	0.2 ± 0.010 ^b^

Note: Values with different lowercase letters in superscript in the same line were significantly different at *p* < 0.05 by LSD in ANOVA.

**Table 2 foods-14-01868-t002:** The relative odor activity values (ROAV ≥ 1) and related information for four rose beverage samples.

Compounds	CAS	Odor Description	Threshold (mg/kg)	OAV			
				LSFR	LFR	SFR	UFR
4−Allyl−2−Methoxyphenol	97−53−0	Strong aroma of cloves	0.00	117,381.10	193,529.17	2015.22	31417.69
(R)−3,7−Dimethyl−6−Octen−1−Ol	106−22−9	Rose	0.01	18,876.00	23,610.79	935.95	12790.26
3,7−Dimethylocta−1,6−Dien−3−Ol	78−70−6	Lilies of the valley, wood, rose, and fruity	0.00	17,389.15	36,328.51	1178.91	18379.08
3,7−Dimethyl−6−Octen−1−Ol	1117−61−9	Rose	0.01	14,921.49	24,945.78	868.02	11672.07
(E)−3,7−Dimethylocta−2,6−Dien−1−Ol	106−25−2	Sweet rose aroma	0.01	13,456.40	15,013.88	58.74	8357.00
Phenylethyl Alcohol	60−12−8	Floral, rosy, honey	0.05	2559.54	5029.25	2810.90	499.52
7−Methyl−3−Methyleneocta−1,6−Diene	123−35−3	Sweet citrus	0.02	2144.04	1872.15	35.87	1805.65
Nonan−1−Ol	143−08−8	Rose, citrus	0.00	1716.28	2594.72	—	131.15
2,4−Bis(1,1−Dimethylethyl)Phenol	96−76−4	Phenolic	0.03	1436.80	2289.11	263.22	851.26
2−Methoxy−2−methylbutane	994−05−8	Camphor	0.02	1092.30	27.48	1760.40	16.64
3−Methylbutyl Acetate	123−92−2	Banana	0.00	755.66	104.92	5617.35	99.04
(4R)−4−Isopropenyl−1−Methylcyclohexene	5989−54−8	Fresh orange aroma	0.03	449.31	343.20	19.09	288.16
Nonanal	124−19−6	Fatty, oily	0.00	405.60	395.93	93.82	1762.02
2−Phenylethyl acetate	103−45−7	Sweet rose aroma, fruity	0.10	361.50	104.82	57.94	72.80
Ethyl hexanoate	123−66−0	Fruity, wine	0.00	303.98	—	21991.37	—
2−(Methylamino)Benzoic Acid Methyl Ester	85−91−6	Mild, slightly sweet	0.02	224.69	520.52	199.08	497.03
1−Methyl−4−(1−Methylethenyl)Cyclohexene	555−10−2	Pepper and citrus aroma	0.04	182.44	143.30	5.69	114.21
(2Z)−3,7−Dimethylocta−2,6−Dien−1−Ol	106−25−2	Rose and orange aroma	0.08	175.04	117.53	2.84	34.58
Undecan−2−Ol	1653−30−1	Citrus fruity aroma	0.07	165.33	300.89	15.31	1.37
Methyl 2−Hydroxybenzoat	118−61−6	Holly	0.06	67.80	74.13	2.06	12.71
α−Terpineol	98−55−5	Fresh, green, woody	0.30	51.44	62.64	10.96	26.00
4−Allyl−1,2−Dimethoxybenzene	93−15−2	Aroma of cloves	1.25	39.99	47.50	5.43	48.76
(Z)−3,7−Dimethyl−2,6−octadienylethanoate	105−87−3	Sweet, floral, slightly fruity	0.10	32.96	—	—	2.04
(5S)−5−Isopropyl−2−Methylcyclohexa−1,3−Diene	2243−33−6	Green, woody, peppery notes	0.20	27.46	20.39	<1	19.29
Nonan−2−Ol	628−99−9	Fruity	0.28	17.20	34.50	1.37	3.78
Decan−1−Ol	112−30−1	Rose and orange flower aroma	0.20	12.25	4.40	1.20	—
Hexan−1−Ol	111−27−3	Fruity	0.20	6.54	11.56	<1	<1
Octanoic Acid	124−07−2	Sweat, cheese	5.00	3.23	<1	3.24	<1
Nonanoic Acid	112−05−0	Unpleasant smell	1.50	3.10	5.00	—	<1
Benzaldehyde	100−52−7	Almond, burnt sugar	0.30	2.04	3.43	<1	18.89
Benzyl Alcohol	100−51−6	Floral, sweet	5.50	1.73	2.83	—	—
Ethyl Octanoate	106−32−1	Pineapple	0.65	1.45	<1	75.37	<1
5−Isopropyl−2−Methylphenol	89−83−8	Musk, grass, spices	1.50	<1	2.74	<1	1.46
(E)−β−Ionone	14901−07−6	Pleasant, sweet citrus	0.47	—	<1	<1	4.27

## Data Availability

The original contributions presented in the study are included in the article/Appendix A; further inquiries can be directed to the corresponding author.

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
