# Peer review of "Influence of Lactiplantibacillus plantarum and Saccharomyces cerevisiae Individual and Collaborative Inoculation on Flavor Characteristics of Rose Fermented Beverage"

_foods, 2025, doi:10.3390/foods14111868_

Round 1
Reviewer 1 Report
Comments and Suggestions for Authors
General Comments
This manuscript investigates the effect of different starter culture combinations on the flavor characteristics of rose-fermented beverages. I appreciate the focus on biotransformation as a means to enhance the nutritional and sensorial quality of fermented foods and beverages. However, I believe the depth of investigation in this article regarding the mechanistic basis of biotransformation needs to be strengthened, particularly since the flavor changes are hypothesized to originate from microbial activity. Below are my major and minor comments, primarily emphasizing factual accuracy, experimental design clarity, and data transparency.
Major Comments
- Theoretical sugar requirement for microbial growth:
The experimental design aims to evaluate the influence of different starter cultures on physicochemical and flavor properties. However, microbes must be metabolically active before fermentation begins. In Table 1, the initial sugar concentration is reported as 36 g/L, which appears insufficient to support meaningful microbial growth and metabolic activity.
- How was the required sugar amount calculated to support just cell growth, prior to flavor compound production?
- What source or literature supports this sugar requirement?
- Please explain whether a growth medium or adaptation step was conducted prior to fermentation.
- Unclear microbial enumeration prior to fermentation:
In Line 107, the manuscript states that "5–7 log CFU/mL co-culture" was used. However, it is unclear how the authors balanced the two strains in co-culture prior to fermentation. - Inconsistency between substrate and organic acid yield:
In Line 250, the authors report that LFR and LSFR produced 23 g/L and 6.29 g/L of organic acids, respectively. However, according to Table 1, the initial total sugar concentration was only 0.36 g/L. This raises a significant concern regarding the source of carbon for such high levels of organic acid production. The authors are requested to clarify how this yield was achieved given the limited carbohydrate substrate. Were there additional carbon sources present that were not reported? A detailed explanation is necessary to reconcile this discrepancy. - What are the minimum taste threshold values for each amino acid used to support the claims made in Lines 239–245 and Figure 1B? Based on the concentrations shown in Figure 1, most amino acids do not appear to reach the threshold required for a TAV > 1 across all fermentation conditions. A similar issue applies to the organic acids discussed in Lines 266–274 and Figures 1C–D. For the sake of reproducibility and transparency, I strongly recommend including a table listing the taste threshold concentration of each compound (amino acids and organic acids), along with appropriate literature references used in the TAV calculation.
Minor Comments
- Introduction Section:
The introduction does not sufficiently highlight the specific knowledge gap this research addresses. Please clarify what is novel in this study compared to prior work on fermented floral beverages or starter culture effects. - Figure 2 Legend:
The term “probiotics” is used, but there is no evidence provided (e.g., in vitro or in vivo assays) that the strains used meet probiotic criteria. Please justify or revise the terminology. - Line 330–331:
The claim that LSFR is not significantly different from others contradicts Figure 3B, where LSFR appears significantly different. Please ensure alignment between text and figures. - Line 167:
Please specify the number of biological and technical replicates used in the experiment. - Lines 169–170:
The statistical analysis section lacks detail. Please state:
- Which post-hoc test was used
- Whether assumptions for ANOVA (normality, homogeneity) were verified
Author Response
- Response to comments of Reviewer #1 (highlighted with yellow color)
Comment 1: Theoretical sugar requirement for microbial growth:
The experimental design aims to evaluate the influence of different starter cultures on physicochemical and flavor properties. However, microbes must be metabolically active before fermentation begins. In Table 1, the initial sugar concentration is reported as 36 g/L, which appears insufficient to support meaningful microbial growth and metabolic activity.
- How was the required sugar amount calculated to support just cell growth, prior to flavor compound production?
- What source or literature supports this sugar requirement?
- Please explain whether a growth medium or adaptation step was conducted prior to fermentation.
Response:Accept. Thank you for the constructive suggestions put forward by the reviewers. We checked the original data and found that there was a mistake in the data calculation. To further answer the reviewers' questions, we remeasured the total sugar content, and the results confirmed that the initial sugar content in the rose beverage should be 36.52g/L. We are very sorry for such a mistake. We have corrected it in the main text in Line 229-231.
Comment 2: Unclear microbial enumeration prior to fermentation:
In Line 107, the manuscript states that "5–7 log CFU/mL co-culture" was used. However, it is unclear how the authors balanced the two strains in co-culture prior to fermentation.
Response:The bacterial cells were resuspended with sterilized physiological saline, and the bacterial concentration was adjusted to 8 log CFU/mL. The two bacterial solutions were mixed in equal proportions, and then inoculated in proportion. In Table 1, we added the count of live bacteria inoculated at 0 hours and 24 hours.
Comment 3: Inconsistency between substrate and organic acid yield:
In Line 250, the authors report that LFR and LSFR produced 23 g/L and 6.29 g/L of organic acids, respectively. However, according to Table 1, the initial total sugar concentration was only 0.36 g/L. This raises a significant concern regarding the source of carbon for such high levels of organic acid production. The authors are requested to clarify how this yield was achieved given the limited carbohydrate substrate. Were there additional carbon sources present that were not reported? A detailed explanation is necessary to reconcile this discrepancy.
Response:Accept. Please refer to the answer to Question 1.
Comment 4: What are the minimum taste threshold values for each amino acid used to support the claims made in Lines 239–245 and Figure 1B?
Based on the concentrations shown in Figure 1, most amino acids do not appear to reach the threshold required for a TAV > 1 across all fermentation conditions. A similar issue applies to the organic acids discussed in Lines 266–274 and Figures 1C–D. For the sake of reproducibility and transparency, I strongly recommend including a table listing the taste threshold concentration of each compound (amino acids and organic acids), along with appropriate literature references used in the TAV calculation.
Response:Accept. We have checked the original data and found that there was an error in the use of the unit of the threshold. In this version of the revised draft, we corrected the results in Figure 1 and Line 292-295. Meanwhile, we also submitted an attached table, which contained the TAVs calculation results, taste threshold and references.
Comment 5: Introduction Section: The introduction does not sufficiently highlight the specific knowledge gap this research addresses. Please clarify what is novel in this study compared to prior work on fermented floral beverages or starter culture effects.
Response:Accept. We have rewritten the second paragraph in the introduction and introduced the research situation of predecessors in Line 44-66.
Comment 6: Figure 2 Legend: The term “probiotics” is used, but there is no evidence provided (e.g., in vitro or in vivo assays) that the strains used meet probiotic criteria. Please justify or revise the terminology.
Response:Accept. We have rewritten the second paragraph in the introduction and introduced the research situation of predecessors in Line 44-66.
Comment 7: Line 330–331: The claim that LSFR is not significantly different from others contradicts Figure 3B, where LSFR appears significantly different. Please ensure alignment between text and figures.
Response:Accept. We have accordingly revised the result in Page 9, Line 345-354.
Comment 8: Line 167: Please specify the number of biological and technical replicates used in the experiment.
Response:Accept. We have added the number of biological and technical replicates used in the experiment in Page 6, Line 213.
Comment 9: Lines 169–170: The statistical analysis section lacks detail. Please state:
- Which post-hoc test was used
- Whether assumptions for ANOVA (normality, homogeneity) were verified
Response:Accept. We have added the details of statistical analysis section in Page 6, Line 2177-221.
Reviewer 2 Report
Comments and Suggestions for Authors
Publication Influence of Lactiplantibacillus plantarum and Saccharomyces cerevisiae individual and collaborative inoculation on flavor characteristics of rose fermented beverage reported interesting results from different fermentation schemes of rose beverages. The results observes how different fermentation achemes effects on the total concentrations of phenolic compounds, free amino acids, and volatile compounds. In addition, authors have used E-nose in their study. The publication has potential to be published in Foods after major revision.
Abstract: L22-23 you say that you revealed significant difference in volatile flavor compounds with E-nose. However, in L296 you say “SFR was significantly different from the other samples” and in L299 you say “The E-nose technique alone was insufficient to analyze the flavor characteristics of the three fermented rose beverages.”. This is inconsistent with the abstract. Furthermore, how did you calculate the significant differences? If you did not, you should modify how you state these results.
Materials and methods
Sample preparation: Please specify which parts of rose you used.
Fermentation: Did you do any biological fermentation replicates? If so, add this to the sample preparations. If not, please explain why you chose not to do them. Biological replicates are crucial in research done with fermentations, because you can get different results between fermentations. With replicated fermentations you can average your results to correspond more accurately reality.
Determination of free amino acids (FAAs) and organic acids: Specify what kind of membranes were used (L115 and 120). In addition, specify which kind of chromatographic column was used in separation (L115).
Extraction of VOCs using HS-SPME: Specify which kind of fibre was used to extract the volatile compounds.
Qualitative and quantitative analysis: Please take account that SPME cannot be used to calculate accurate concentration, but it can be used for semi-qualitative concentrations. Please revise this section according to this.
Method descriptions for color analysis, total phenolic, total flavonoid, and reducing sugars analyses are missing from the Methods section. In addition, add how you determined TAV and OAV. Reducing sugars are not discussed at any point, add this to the manuscript. Furthermore, you should add something about TAV and OAV to Introduction to help reader to understand their significance and what they are based on. Did you calculated OAV based on other studies detection thresholds? This kind of things should be addressed and explained in the publication, because this approach does not give as accurate results as you would had done it by yourselves. In addition, SPME methods are not recommended to use with OAV, because you cannot get accurate concentrations with it. These things you should take account in your manuscript too.
Figure captions: All of your figure captions, expect Figure 1, are not specific enough. Please specify, certain aspects, for example, sample and variable amounts in PCAs and PLS-DA (Figure 2, 3, and 6), what means A, B, C, etc. in Figure 4, add statistically significant difference in radar plot (figure 2). Caption of Figure 5 (L475-476) is not a caption at all, revise it. In addition, the texts in Figure 4 are illegibly.
Table captions: Table 2 modify the caption (L469) to be more informative. Do not start with “presents” and highlight that it is about the compounds.
General comments
L88: Specify what the physicochemical indexes means.
L89: You give an abbreviation for rose-based functional beverage, but you never use it. Use it or remove the abbreviation from here. Same with the “Picp-2” and “SY” (L90), only usen in Methods, where they are understandable, but are they necessary here?
L203-204: add reference.
L239: It is not common to use phrases such as “We calculated”. Please revise this and other part you have used this phrase (L266).
L264: L. plantarum in italics.
L298: Add Figure reference after “PCA map”.
Comments on the Quality of English Language
There is a major problem with English language. In the Introduction the tense is wrong. In addition, sentence structures are and word orders should be carefully checked and modified. Through out textthe manuscript, you use “rose functional beverage”, which would be more correct as “functional rose beverage”, please revise this issue as well.
Author Response
Response to comments of Reviewer #2 (highlighted with blue color)
Comment 1: Abstract: L22-23 you say that you revealed significant difference in volatile flavor compounds with E-nose. However, in L296 you say “SFR was significantly different from the other samples” and in L299 you say “The E-nose technique alone was insufficient to analyze the flavor characteristics of the three fermented rose beverages.”. This is inconsistent with the abstract. Furthermore, how did you calculate the significant differences? If you did not, you should modify how you state these results.
Response:Accept. We have accordingly revised the result in Page 9, Line 345-354 and Figure 2.
Comment 2: Materials and methods: Sample preparation: Please specify which parts of rose you used.
Response:Accept. We have added the details in Page 4, Line 115.
Comment 3: Fermentation: Did you do any biological fermentation replicates? If so, add this to the sample preparations. If not, please explain why you chose not to do them. Biological replicates are crucial in research done with fermentations, because you can get different results between fermentations. With replicated fermentations you can average your results to correspond more accurately reality.
Response:Accept. We have added the number of biological and technical replicates used in the experiment in Page 6, Line 213.
Comment 4: Determination of free amino acids (FAAs) and organic acids: Specify what kind of membranes were used (L115 and 120). In addition, specify which kind of chromatographic column was used in separation (L115).
Response:Accept. We have added the details about the type of membrane and the model of the chromatographic column in Page 5, Line 154-161.
Comment 5: Extraction of VOCs using HS-SPME: Specify which kind of fibre was used to extract the volatile compounds.
Response:Accept. We have added the details about the type of fiber in Page 5, Line 172-173.
Comment 6: Qualitative and quantitative analysis: Please take account that SPME cannot be used to calculate accurate concentration, but it can be used for semi-qualitative concentrations. Please revise this section according to this.
Response:Accept. We have accordingly revised in Page 5, Line 198.
Comment 7: Method descriptions for color analysis, total phenolic, total flavonoid, and reducing sugars analyses are missing from the Methods section. In addition, add how you determined TAV and OAV. Reducing sugars are not discussed at any point, add this to the manuscript. Furthermore, you should add something about TAV and OAV to Introduction to help reader to understand their significance and what they are based on. Did you calculated OAV based on other studies detection thresholds? This kind of things should be addressed and explained in the publication, because this approach does not give as accurate results as you would had done it by yourselves. In addition, SPME methods are not recommended to use with OAV, because you cannot get accurate concentrations with it. These things you should take account in your manuscript too.
Response:Accept. We have accordingly revised in Page 4, Line 123-148, Page 5-6, Line 166-167, Line 201-206, respectively.
Comment 8: Figure captions: All of your figure captions, expect Figure 1, are not specific enough. Please specify, certain aspects, for example, sample and variable amounts in PCAs and PLS-DA (Figure 2, 3, and 6), what means A, B, C, etc. in Figure 4, add statistically significant difference in radar plot (figure 2). Caption of Figure 5 (L475-476) is not a caption at all, revise it. In addition, the texts in Figure 4 are illegibly.
Response:Accept. We have re-edited all the figure captions.
Comment 9: Table captions: Table 2 modify the caption (L469) to be more informative. Do not start with “presents” and highlight that it is about the compounds.
Response:Accept. We have revised the caption of Table 2.
Comment 10: General comments:
L88: Specify what the physicochemical indexes means.
Response:Accept. We have accordingly revised in Page 2, Line 93-94.
L89: You give an abbreviation for rose-based functional beverage, but you never use it. Use it or remove the abbreviation from here. Same with the “Picp-2” and “SY” (L90), only usen in Methods, where they are understandable, but are they necessary here?
Response:Accept. We have accordingly revised in Page 2, Line 95.
L203-204: add reference.
Response:We have added the reference in Page 6, Line 237.
L239: It is not common to use phrases such as “We calculated”. Please revise this and other part you have used this phrase (L266)
Response:Accept. We have accordingly revised in Page 8, Line 292-295, Line 315-316.
L264: L. plantarum in italics.
Response:Accept.
L298: Add Figure reference after “PCA map”.
Response:Accept.
Comment 11: Comments on the Quality of English Language: There is a major problem with English language. In the Introduction the tense is wrong. In addition, sentence structures are and word orders should be carefully checked and modified. Throughout text the manuscript, you use “rose functional beverage”, which would be more correct as “functional rose beverage”, please revise this issue as well.
Response:Accept. We have checked the English language of the entire text and made corrections to the inappropriate words used.
Reviewer 3 Report
Comments and Suggestions for Authors
In this work, the authors discuss about the Influence of L. plantarum and S. cerevisiae (individual and collaborative inoculation) on flavor characteristics of rose fermented beverage. The topic is interesting and the manuscript has a potential from a scientific point of view. However, there are a few points that require the authors’ attention.
Introduction is in general well-written. Only one suggestion:
Lines 69-72: In my opinion, since the experimental part relies on co-inoculations, the authors should provide a few examples of studies using mixed strain cultures that enhance aroma and odor in fruit-based fermented beverages (e.g. https://doi.org/10.1002/jsfa.10363, etc).
Materials and methods is well organized:
Line 103: Why did the authors choose this temperature (70 C) for extracting the dried rose? Were any preliminary experiments included? It should be explained.
Lines 105-106: The preparation and growth conditions of these strains (for inoculation) should be better described. If needed, create a new subsection.
Lines 128-159 could be merged as one subsection, since all refer to HS-SPME GC-MS analysis.
The authors should provide appropriate references for each subsection of the materials and methods. If however these methods were solely developed for this study, then the authors should provide calibration curves and limits of detection and quantification (especially for instrumental analysis) as supplementary material.
Results & Discussion:
The discussion is in general well-organized and pretty detailed in certain parts. The potential influence of fermentation duration and other important factors (for example pH, temperature, etc) on the volatiles formations and concentrations should be better discussed. At this point it is somehow under-represented, so please edit.
The authors keep referring to the influence of different probiotic microorganisms on the aroma compounds of rose fermented beverages (in various subsections). However, the actual populations of the microorganisms in the final products are not presented. I believe this is crucial for this kind of work, since the authors have dedicated a great part of their introduction discussing about probiotics. Nevertheless, if manufacturing of a probiotic drink is not part of the rationale of this study, then the introduction (especially lines 44-58) should be extensively edited in order to focus on fermentation-derived aroma compounds and mixed starter cultured and not probiotic microorganisms.
Again, the authors describe in detail the importance of their results in correlation to the sensory characteristics, measured by GC/MS, electronic nose, etc. This is great work and very detailed. However, the lack of sensory trial results is disappointing and a clear disadvantage. In order to overcome this fact, I can only recommend the authors to create a final paragraph in the discussion section that should contain and highlight all important aspects of this manuscript. The authors should expand to the technological, market/economical, environmental, health, nutritional, etc insights gained from this work that could suggest scale up and maybe industrial application of the rose fermented beverage.
Conclusions are ok. Slight editing might me required, based on the comments above.
Author Response
- Response to comments of Reviewer #3 (highlighted with purple color)
Comment 1: Introduction is in general well-written. Only one suggestion:
Lines 69-72: In my opinion, since the experimental part relies on co-inoculations, the authors should provide a few examples of studies using mixed strain cultures that enhance aroma and odor in fruit-based fermented beverages (e.g. https://doi.org/10.1002/jsfa.10363, etc).
Response:Accept. In the fourth paragraph of the introduction, we mentioned that co-fermentation with Lactobacillus plantrum and Saccharomyces cerevisiae can enhance the flavor, and cited the literature in Line 103-105.
Comment 2: Materials and methods is well organized:
Line 103: Why did the authors choose this temperature (70 C) for extracting the dried rose? Were any preliminary experiments included? It should be explained.
Response:Temperature promotes the dissolution of nutrients, but excessively high temperatures can destroy the flavor profiles and nutritional components of rose beverages. Before conducting the formal experiment, we screened the temperature of the water extraction. It was found that under the treatment condition of 70℃, the rose extract had the most intense fragrance and the best nutritional indicators such as amino acids, polyphenols and flavonoids. Therefore, we adopted this temperature for the subsequent experiments.
Lines 105-106: The preparation and growth conditions of these strains (for inoculation) should be better described. If needed, create a new subsection.
Response:Accept. We have added the culture method of the strain to the materials and methods in Line 108-122.
Lines 128-159 could be merged as one subsection, since all refer to HS-SPME GC-MS analysis.
Response:Accept.
The authors should provide appropriate references for each subsection of the materials and methods. If however these methods were solely developed for this study, then the authors should provide calibration curves and limits of detection and quantification (especially for instrumental analysis) as supplementary material.
Response:Accept. We have supplemented the specific analytical methods in the material approach.
Comment 3: Results & Discussion:
The discussion is in general well-organized and pretty detailed in certain parts. The potential influence of fermentation duration and other important factors (for example pH, temperature, etc) on the volatiles formations and concentrations should be better discussed. At this point it is somehow under-represented, so please edit.
Response:Accept. We have accordingly added the paragraphs for discussion in Page 12, Line 500-507.
The authors keep referring to the influence of different probiotic microorganisms on the aroma compounds of rose fermented beverages (in various subsections). However, the actual populations of the microorganisms in the final products are not presented. I believe this is crucial for this kind of work, since the authors have dedicated a great part of their introduction discussing about probiotics. Nevertheless, if manufacturing of a probiotic drink is not part of the rationale of this study, then the introduction (especially lines 44-58) should be extensively edited in order to focus on fermentation-derived aroma compounds and mixed starter cultured and not probiotic microorganisms.
Response:Accept. We have rewritten the second paragraph in the introduction and introduced the research situation of predecessors in Line 44-66.
Again, the authors describe in detail the importance of their results in correlation to the sensory characteristics, measured by GC/MS, electronic nose, etc. This is great work and very detailed. However, the lack of sensory trial results is disappointing and a clear disadvantage. In order to overcome this fact, I can only recommend the authors to create a final paragraph in the discussion section that should contain and highlight all important aspects of this manuscript. The authors should expand to the technological, market/economical, environmental, health, nutritional, etc insights gained from this work that could suggest scale up and maybe industrial application of the rose fermented beverage.
Response:Accept. We have accordingly added the paragraphs for discussion in Line 571-593.
Conclusions are ok. Slight editing might me required, based on the comments above.
Response:Accept.
Round 2
Reviewer 1 Report
Comments and Suggestions for Authors
The authors have made appropriate revisions in response to the feedback, and I am satisfied with the current version.
Author Response
Comment 1: The authors have made appropriate revisions in response to the feedback, and I am satisfied with the current version.
Response:Accept.
Reviewer 2 Report
Comments and Suggestions for Authors
Authors has improved the manuscript Influence of Lactiplantibacillus plantarum and Saccharomyces cerevisiae individual and collaborative inoculation on flavor characteristics of rose fermented beverage by answering reviewers’ comments. However, there is still some issues to address.
First of all, the English language still needs improvement in the Introduction and the text needs careful revision, because there are some errors throughout.
Supplementary material: Table S1. has different naming systems, for example, ethyl hexanoate and ethyl caprylate belongs to two different system. Revise the compound names. In addition, authors have not indicated that the concentrations are semi-quantitative, revise this too.
L132-134: How did you measure trichromatic value and with what instrument?
Determination of total phenols and total flavonoids both are missing calibration curve information.
L201: You should write out here the meaning of ROAV.
L278: Add “respective” between “the” and “proportions”.
L341: When you are talking about the samples in PCAs call the samples not points. Revise these throughout the text.
Figure 2 caption L352: Add the information telling is the plot Scores or Loadings plot.
L359: HS-SPME-GC-MS
L376-378: Add how many volatile compounds you used to construct the PCA.
L565-587: Why this paragraph is here not in Conclusions?
Comments on the Quality of English Language
English language should be improved in the Introduction.
Author Response
Authors has improved the manuscript Influence of Lactiplantibacillus plantarum and Saccharomyces cerevisiae individual and collaborative inoculation on flavor characteristics of rose fermented beverage by answering reviewers’ comments. However, there is still some issues to address.
Comment 1:First of all, the English language still needs improvement in the Introduction and the text needs careful revision, because there are some errors throughout.
Response: Accept. We have enhanced the English language of the entire text.
Comment 2:Supplementary material: Table S1. has different naming systems, for example, ethyl hexanoate and ethyl caprylate belongs to two different system. Revise the compound names. In addition, authors have not indicated that the concentrations are semi-quantitative, revise this too.
Response: Accept. We have accordingly revised the compound names. And we also changed “Content (ug/mL)” to “Relative content (ug/mL)”.
Comment 3:L132-134: How did you measure trichromatic value and with what instrument?
Determination of total phenols and total flavonoids both are missing calibration curve information.
Response: Accept. We measured trichromatic value using the CIELAB model (UV–vis 2450, Shimadzu Corporation, Kyoto, Japan), and we also added the calibration curve for determination of total phenols and total flavonoids.
Comment 4:L201: You should write out here the meaning of ROAV.
Response: Accept.
Comment 5:L278: Add “respective” between “the” and “proportions”.
Response: Accept.
Comment 6:L341: When you are talking about the samples in PCAs call the samples not points. Revise these throughout the text.
Response: Accept.
Comment 7:Figure 2 caption L352: Add the information telling is the plot Scores or Loadings plot.
Response: Accept. We have added the plot scores in caption of Figure 2.
Comment 8:L359: HS-SPME-GC-MS
Response: Accept.
L376-378: Add how many volatile compounds you used to construct the PCA.
Response: Accept.
L565-587: Why this paragraph is here not in Conclusions?
Response: The main purpose of this section is to discuss the innovations and shortcomings of our research, as well as the in-depth work that needs to be carried out in the future. It is not the achievement of this research, so we have placed it in the discussion part.
Reviewer 3 Report
Comments and Suggestions for Authors
The authors have addressed the majority of my comments and the manuscript has been substantially improved. There are only a few minor details remaining:
1) Lines 115-116: Add a sentence here or in the discussion section about the reasons responsible for choosing 70 degrees C for the extraction.
2) Appropriate references are required in each subsection of the materials and methods. This is important for clarity reasons and for better repeatability. No references at all were found (in the materials and methods) at the current manuscript.
3) Remove the word "probiotic" in lines 20, 349 and 596. Use "strains" or "microorganisms" instead.
Author Response
The authors have addressed the majority of my comments and the manuscript has been substantially improved. There are only a few minor details remaining:
Comment 1: Lines 115-116: Add a sentence here or in the discussion section about the reasons responsible for choosing 70 degrees C for the extraction.
Response:Accept. We have accordingly revised in Page 4, Line 115-116.
Comment 2: Appropriate references are required in each subsection of the materials and methods. This is important for clarity reasons and for better repeatability. No references at all were found (in the materials and methods) at the current manuscript.
Response:Accept. We have added references to the materials and methods section.
Comment 2: Remove the word "probiotic" in lines 20, 349 and 596. Use "strains" or "microorganisms" instead.
Response:Accept. We have made corresponding modifications.